# Metabolite Ratios as Quality Indicators for Pre-Analytical Variation in Serum and EDTA Plasma

**DOI:** 10.3390/metabo11090638

**Published:** 2021-09-18

**Authors:** Sven Heiling, Nadine Knutti, Franziska Scherr, Jörg Geiger, Juliane Weikert, Michael Rose, Roland Jahns, Uta Ceglarek, André Scherag, Michael Kiehntopf

**Affiliations:** 1Institute of Clinical Chemistry and Laboratory Diagnostics and Integrated Biobank Jena (IBBJ), University Hospital Jena, Am Klinikum 1, 07747 Jena, Germany; Nadine.Knutti@med.uni-jena.de (N.K.); Franziska.Scherr@med.uni-jena.de (F.S.); Michael.Rose@med.uni-jena.de (M.R.); 2Interdisciplinary Bank of Biological Material and Data Würzburg (IBDW), Straubmühlweg 2a, Haus A9, 97078 Würzburg, Germany; geiger_j1@ukw.de (J.G.); jahns_r@ukw.de (R.J.); 3Institute of Laboratory Medicine, Clinical Chemistry and Molecular Diagnostics, University Hospital Leipzig, 04103 Leipzig, Germany; Juliane.Weikert@medizin.uni-leipzig.de (J.W.); Uta.Ceglarek@medizin.uni-leipzig.de (U.C.); 4LIFE Leipzig Research Center for Civilization Diseases, University of Leipzig, 04103 Leipzig, Germany; 5Institute of Medical Statistics, Computer and Data Sciences, Jena University Hospital, Bachstrasse 18, 07743 Jena, Germany; Andre.Scherag@med.uni-jena.de

**Keywords:** quality indicators, biomarker, hypoxanthine, inosine, guanosine, eicosanoids, time-to-centrifugation, pre-analytical variation

## Abstract

In clinical diagnostics and research, blood samples are one of the most frequently used materials. Nevertheless, exploring the chemical composition of human plasma and serum is challenging due to the highly dynamic influence of pre-analytical variation. A prominent example is the variability in pre-centrifugation delay (time-to-centrifugation; TTC). Quality indicators (QI) reflecting sample TTC are of utmost importance in assessing sample history and resulting sample quality, which is essential for accurate diagnostics and conclusive, reproducible research. In the present study, we subjected human blood to varying TTCs at room temperature prior to processing for plasma or serum preparation. Potential sample QIs were identified by Ultra high pressure liquid chromatography tandem mass spectrometry (UHPLC-MS/MS) based metabolite profiling in samples from healthy volunteers (n = 10). Selected QIs were validated by a targeted MS/MS approach in two independent sets of samples from patients (n = 40 and n = 70). In serum, the hypoxanthine/guanosine (HG) and hypoxanthine/inosine (HI) ratios demonstrated high diagnostic performance (Sensitivity/Specificity > 80%) for the discrimination of samples with a TTC > 1 h. We identified several eicosanoids, such as 12-HETE, 15-(S)-HETE, 8-(S)-HETE, 12-oxo-HETE, (±)13-HODE and 12-(S)-HEPE as QIs for a pre-centrifugation delay > 2 h. 12-HETE, 12-oxo-HETE, 8-(S)-HETE, and 12-(S)-HEPE, and the HI- and HG-ratios could be validated in patient samples.

## 1. Introduction

Access to high-quality, well characterized human biological samples is crucial for accurate diagnostics in the clinical routine laboratory and for reliable, conclusive biomedical research. In particular, omics technologies rely on high-quality samples obtained by standardized pre-analytical workflows [1].

In practice, sampling procedures and pre-analytical conditions vary substantially in different clinical settings. Similarly, relevant information on these pre-analytical conditions is often missing or insufficient for assessing sample quality. Missing information on pre-analytical conditions may be approximated by retrospective analyses of specific biomarkers known to be affected by distinct pre-analytical conditions. Metabolomics appears particularly suited to identifying potential quality indicators (QI), as metabolites are highly prone to pre-analytical variations. Nevertheless, only a few potential QIs have been described for critical steps of sample processing, such as time-to-centrifugation (TTC) [2,3,4,5,6,7], time-to-freeze [8,9,10,11], or temperature variation [3,4,5]. Examples of well-characterized QIs for TTC are (4E,14Z)-sphingadienine-C-18-1-phosphate (S1P-d18:2) [12], glucose, lactate, arginine, ornithine [3,6] and taurine [7,13]. Particularly, S1P-d18:2, tested in more than 1400 randomly selected serum and plasma samples during single- and multicenter trials in 11 biobanks across three countries, has been proven useful as a QI [12]. QIs specifically reflecting prolonged TTF are lysophosphatidylcholines (lysoPCs) [10] and several proteolytic fragments of endogenous peptides such as fibrinogen alpha and beta chains [9,11]. Further studies identified QIs reflecting time delays between blood collection and centrifugation, variations in time to freeze, and ambient temperature. In a previous study, we have shown that a TTC-dependent increase in taurine can be used in both healthy and diseased individuals to distinguish high qualitative biomaterials from samples undergoing a pre-centrifugation delay of more than one hour [13]. TTC related increase of taurin was temperature dependent as also shown for time-dependent changes of other metabolites, for example, 15-hydroxyeicosatetraenoic acid (15-HETE), glutamate, and cysteine [4]. The largest changes in the metabolome were observed before cell separation, depending on duration and temperature [5,14]. Consequently, the TTC should be minimized and elevated ambient temperature avoided.

While QIs usually respond very clearly to a prolonged TTC or TTF, several studies have shown that their interindividual variability is in some cases greater than the changes observed as a result of altered preanalytics [10,15,16]. Hence, general cutoff thresholds for individual metabolites seem to be insufficient in terms of pre-analytical sample quality assessment. Using ratios of metabolites instead of absolute metabolite concentrations might help to increase the robustness of QIs. Metabolite ratios for ornithine/arginine [6], lactate/glucose [5], and total lysophosphatidylcholines/total phosphatidylcholines [10] have thus been suggested for assessing sample quality.

Here, we used a targeted metabolite profiling approach to discover biomarkers suitable as QIs for the pre-analytical phase, in particular the time-to-centrifugation for both human serum and EDTA plasma. In samples from healthy individuals (n = 10), we could identify novel, and confirm already reported, metabolite ratios that might be well suited as pre-analytical QIs. Among the potential candidates are nucleosides (hypoxanthine/inosine, hypoxanthine/guanosine), amino acids (ornithine/arginine), and eicosanoids. The QI candidates were confirmed by targeted MS/MS analysis in two independent reference sample sets from healthy volunteers (*validation sample 1*—n = 10, *validation sample 2*—n = 23) and validated in two independent sample sets from patients with rheumatic and cardiovascular diseases (*validation sample 3*—n = 40, *validation sample 4*—serum n = 70; EDTA plasma n = 49).

## 2. Results

### 2.1. Targeted Metabolite Profiling of Serum and EDTA Plasma Samples from Healthy Individuals for TTC Quality Indicator Discovery

Targeted metabolite profiling was performed in a pilot study with samples from a set of healthy individuals (*discovery sample*, n = 10) to compare human serum and EDTA plasma processed after 0.5 and 2 h pre-centrifugation delays. Altogether, 752 serum and 714 plasma metabolites were detected. After data pre-processing (see Methods), 674 serum and 632 plasma metabolites were identified that differed between 0.5 h to 2 h TTC. Effects of a prolonged pre-centrifugation delay were more pronounced in serum than in EDTA plasma. In serum and plasma, 10.4% and 3.9% of all metabolites varied between incubation times, respectively.

In serum, 59 metabolites were increased (seven with a fold change >2), and 11 metabolites were decreased (five with a fold change <0.5). The most remarkable changes were observed for several dipeptides, fibrinogen cleavage peptides, nucleosides, and amino acids (Figure 1A, Appendix A). The affected metabolites could be assigned to the following metabolic pathways/subpathways: “glutamate metabolism”, “methionine, cysteine, S-adenosylmethionine (SAM) and taurine metabolism”, “purine metabolism”, “urea cycle”, as well as diverse subpathways of the lipid metabolism (Appendix A). In EDTA plasma, 21 metabolites were increased (one with a fold change >2), and four metabolites were decreased (Figure 2A). In EDTA plasma, changes were observed for endocannabinoids and isolated metabolites of various subpathways, such as “fatty acid metabolism” or “urea cycle” (Appendix A).

Ratios of metabolites that could be affected by enzymatic or non-enzymatic processes involved in the metabolic pathways of respective QIs were calculated. The relationships were visualized by an over-representation analysis of the metabolites significantly affected by TTC, using KEGG (https://www.metaboanalyst.ca, accessed on 30 June 2020) as a pre-defined reference pathway set. Due to the insufficient coverage of the KEGG pathways by the analytes examined, only 42 metabolites in serum and 13 metabolites in EDTA plasma could be included in the analysis. The pathway impact and *p*-values were calculated and plotted (Figure 1B and Figure 2B). As a result of concentration changes of hub metabolites (such as asparagine, aspartate, arginine, glycine, glutamate, serine, and taurine) in serum, the amino acid metabolism was enriched. Enrichment in the purine and sphingolipid metabolic pathways was observed as well. Sphingosine, sphinganine, and sphingosine-1-phosphate were assigned to the sphingolipid metabolism and showed increased levels after 2 h TTC. Within the purine metabolism, we observed an increase in xanthine, hypoxanthine, and glycine, and a decrease in adenosine, cyclic AMP, guanosine, and inosine (Appendix A). Using this information, we calculated metabolite ratios, such as hypoxanthine/inosine (HI-ratio), hypoxanthine/guanosine (HG-ratio,) xanthine/guanosine (XG-ratio), xanthine/inosine (XI-ratio), and ornithine/arginine (OA-ratio) as potential QIs in serum (Appendix A). In EDTA plasma, only the “arginine and ornithine metabolism” was enriched (Appendix A). The ratio of arginine to ornithine was also selected in EDTA plasma (Appendix A).

Further potential QIs were sought by ChemRICH [17] analysis to overcome the limitations of pathway enrichment statistics, such as lacking pathway information for metabolites or metabolites common to different pathways. All metabolites with existing PubChem IDs were included in this analysis. Four hundred and nineteen serum metabolites were grouped into 68 metabolite clusters, of which 19 clusters show differences (Figure 1D). Among these metabolic clusters were saturated fatty acids, ethanolamines, linoleic acids, and dipeptides. Additionally, we observed changes in the OH-FA_17_3_1 cluster, comprising eicosanoids, such as 12-HETE and 15-HETE (Appendix A). Based on these results, metabolites of the eicosanoid pathway were selected for further investigation.

In EDTA plasma, 458 metabolites were grouped in 62 clusters, of which eight clusters showed differences (Figure 2D). The most pronounced changes were observed in saturated lysophosphatidylcholines, ethanolamines, and basic amino acid clusters (Appendix A).

### 2.2. Validation of Candidate QIs for TTC in Serum and EDTA Plasma Samples from Healthy Volunteers

Based on the targeted metabolite profiling and pathway enrichment, we first examined eight potential candidate QIs (xanthine, hypoxanthine, inosine, guanosine, dihydroorotate, N-carbamoylaspartate, 7-methylguanine, allantoin) from the purine and pyrimidine pathway in serum and EDTA plasma samples in a second, independent sample set of healthy volunteers (*validation sample 1*, n = 10). However, we investigated a more extended grid of 0.5, 1, 2 and 4 h TTC (0, only in EDTA-plasma) and used a quantitative MS/MS-based approach. We observed a TTC-dependent increase for xanthine, hypoxanthine, and dihydroorotic acid (Figure 3). Hypoxanthine showed the largest changes after 1, 2 and 4 h compared to 0.5 h pre-centrifugation delay. Of note, dihydroorotic acid showed the strongest increase after 24 h. As expected, the two purine nucleosides guanosine and inosine were sensitive to prolonged pre-processing and decreased after 2 and 4 h relative to the basal concentration at 0.5 h. To achieve robust QIs, we only calculated the ratios of strongly altered metabolites within a specific metabolic pathway. Ratios between hypoxanthine, inosine, xanthine, and guanosine (HI, HG, XI, and XG) were all increased after 2 and 4 h compared to 0.5 h (Figure 3). However, we observed no evidence of alterations in allantoin and 7-methylguanine in serum samples after a 4 h prolonged blood incubation (Appendix A).

In EDTA plasma, only hypoxanthine and dihydroorotic acid increased after 1, 2, and 4 h relative at 0.5 h. Allantoin, 7-methylguanine, inosine, and N-carbamoylaspartate were not altered after 4 h TTC (Appendix A). Guanosine and xanthine concentrations were below the limit of detection in EDTA plasma samples. Unfortunately, the metabolites adenosine and adenosine 3′,5′-cyclic monophosphate [18], which showed a substantial decrease in intensity in the *discovery sample*, were below the detection limit in the serum and EDTA plasma of the *validation samples* and could neither be analyzed nor integrated into the analysis of metabolite ratios.

Based on the results of the chemical similarity enrichment analysis, we investigated eicosanoids in the human serum of healthy individuals (*validation sample 1*). In general, most metabolites were increased after 4 h prolonged blood incubation compared to 0.5 h (Figure 4), except for 12-oxo-ETE, 11,12-DHET, and 14,15-DHET (Figure 4 and Appendix A). The most significant changes were observed for 12-HETE, 15-(S)-HETE, 12-(S)-HEPE, and (±)13-HODE, resulting in fold changes from 1.8 up to 11.2 (Appendix A). The highest discrimination between the different time points was observed for 12-HETE. In EDTA plasma, we quantified 16 eicosanoids, of which only (±)9-HODE and (±)13-HODE were elevated (Appendix A). Eicosanoid concentrations were higher in serum compared to EDTA plasma, with elevated concentrations of TxB2 (15.9 fold), 12-(S)-HHT (7.8 fold), 5-HETE (7 fold), and 12-HETE (29.4 fold) up to 29 fold at 0.5 h.

Additionally, we investigated the effect of prolonged blood incubation (0.5, 1, 2, and 4 h) on arginine and ornithine in the third sample set of healthy volunteers (*validation sample 2*, n = 23). In both serum and EDTA plasma, ornithine concentrations increased and arginine concentrations decreased (Figure 5). While ornithine concentrations were similar, arginine was about ~65% of the basal concentrations in EDTA plasma compared to serum. The OA-ratios were increased in serum and EDTA plasma after 1, 2 and 4 h pre-centrifugation delays.

We calculated candidate-specific cutoff values by analyzing multiple ROCs for all potential QIs (Table 1). Based on these calculations, guanosine (Sen. 83%, Spec. 100%) and hypoxanthine (Sen. 80%, Spec. 86%) were most suitable for discrimination of TTC > 2 h, whereas the HI- (Sen. 86%, Spec. 83%) and HG-ratio (Sen. 80%, Spec. 100%) showed the highest discrimination for TTC > 1 h. Additionally, the eicosanoids ± 13-HODE (Sen. 100%, Spec. 83%), 15(S)-HETE (Sen. 100%, Spec. 100%), 12(S)-HEPE (Sen. 100%, Spec. 100%), 8(S)-HETE (Sen. 100%, Spec. 100%), 12-HETE (Sen. 100%, Spec. 100%) and 12-oxo-ETE (Sen. 100%, Spec. 100%) were shown to predict a prolonged TTC > 2 h in serum with high sensitivity and specificity. Furthermore, when the optimal cutoff for the selected QIs was calculated for EDTA plasma, only the OA-ratio showed good sensitivity and specificity for identification of samples with a prolonged TTC > 1 h (Sen. 73%, Spec. 85%) or TTC > 2 h (Sen. 83%, Spec. 78%).

### 2.3. Validation of Candidate QIs for TTC in Serum and EDTA Plasma Samples from Patients with Rheumatic and Cardiovascular Diseases

To further validate the accuracy of potential candidate QIs, we applied their cutoffs to samples with known TTCs collected from patients with cardiovascular and rheumatic diseases (*validation samples 3* and *4*). Best results to predict samples with a prolonged TTC > 1 h in serum were observed using the HI- and HG-ratio. The calculated median optimal cutoffs of 1.41 (HI-ratio) and 2.06 (HG-ratio) achieved a Sen. of 79% and 100% and a Spec. of 85% and 73%, respectively. Interestingly, the combination of both thresholds, where either one or the other must be exceeded, results in a sensitivity of 100% and a specificity of 92%. Furthermore, several eicosanoids such as Tetranor-12(S)-HETE (Sen. 92%, Spec. 100%), 12(S)-HEPE (Sen. 100%, Spec. 86%), 8(S)-HETE (Sen. 100%, Spec. 86%), 12-HETE (Sen. 92%, Spec. 86%) and 12-oxo-ETE (Sen. 100%, Spec. 86%) were shown to be able to predict a prolonged TTC > 2 h in serum. In EDTA plasma samples, only the HI-ratio (Sen. 81%, Spec. 85%) could be identified as a promising QI for a TTC > 1 h. The AO-ratio could not be confirmed as QI in serum or EDTA plasma. A detailed overview of the results of all potential QIs is shown in Table 2. A summary of all potential quality indicators for further investigation is shown in Table 3.

## 3. Discussion

Delayed sample processing, in particular prolonged time to centrifugation, has a significant impact on the in vivo metabolome in blood derivatives and can substantially affect clinical diagnostics and research [19]. To ensure high quality of human serum and EDTA plasma samples in biobanks, it is essential to minimize TTC and to keep the dwell time of the samples at room temperature as short as possible [14]. However, due to the lack of standardized pre-analytical workflows [1] and quality assurance tools for workflow-validation, sample history is often incomplete or poorly documented and thus, the impact of pre-processing conditions on sample quality is unclear. For this reason, reliable QIs suitable for retrospective assessment of sample history and thus biospecimen quality are needed. While potential QIs reflecting deviations resulting from prolonged TTC have already been identified [5,6,20], their applicability to patients samples has not been sufficiently validated and demonstrated. Consequently, these QIs are not yet routinely used in biobanks workflows. This study describes several potential metabolite ratios or individual metabolites as QIs, identified in healthy cohorts and confirmed in samples from patients with cardiovascular and rheumatic diseases.

### 3.1. HI- and HG-Ratios as Potential Quality Indicators in Serum

After prolonged TTC at room temperature, we observed either an increase or a decrease in certain metabolites of the purine degradation pathway in serum and EDTA plasma. This variation is consistent with previous studies; for instance, Liu and colleagues [21] described an increase of relative hypoxanthine levels in serum after a delay of 4 h prior to processing.

Moreover, it has been shown that hypoxanthine rapidly increases in whole blood after storage at room temperature before centrifugation [22,23]. Hypoxanthine is endogenously produced in erythrocytes [22], and it is well known that guanosine and inosine are converted by the purine nucleoside phosphorylase (PNP—EC 2.4.2.1), an enzyme that is found in red blood cells (RBCs), to guanine and hypoxanthine [24]. Farthing and colleagues recommended determining inosine and hypoxanthine in plasma samples, as the clotting time of approximately 30 min required for a serum sample would allow significant conversion of inosine to hypoxanthine in the collection tube due to PNP activity from RBCs [25]. RBCs also function as ATP storage vesicles [26], therefore by ATP depletion of RBC [27] and AMP degradation, small polar substances, such as inosine and hypoxanthine, are formed [28] which could passively diffuse into serum or plasma. Interestingly, low PNP activity in plasma [29,30] might explain the stable inosine concentration and the less pronounced increase of hypoxanthine in EDTA plasma compared to serum.

Furthermore, we observed an increase of xanthine in serum samples after prolonged TTC, which might be due to further degradation of hypoxanthine [24] by xanthine oxidase (XO—EC 1.17.3.2). Low activity of XO in plasma might be the reason of inferior xanthine concentrations in EDTA plasma samples [29]. Thus, metabolite ratios of the purine metabolism are particularly suitable to detect TTC-dependent conversion and degradation processes due to the high activity of PNP and XO in serum.

According to the results obtained with samples from healthy individuals and their validation in patients with cardiovascular and rheumatic diseases, it can be concluded that serum samples with pre-centrifugation delays ≤ and >1 h can be reliably discriminated based on the HI- and HG-ratios with high sensitivity and specificity. Additionally, we have observed that the calculation of metabolite ratios may increase the performance of QIs compared to the use of their individual concentrations in cohorts of cardiovascular and rheumatic patients (Table 2). This was also shown by Jain and colleagues who demonstrate that combined QIs increase the prognostic value in samples across multiple clinical centers for a processing delay > 2 h [6].

Even though both HI- and HG-ratios provide promising results as QIs for prolonged TTC, components of the purine metabolism can be affected by multiple individual factors such as exercise [31,32], nutrition, or disease. For example, patients with acute cardiac ischemia [28,33] and multiple sclerosis [34] show altered or elevated levels of hypoxanthine and inosine. Hypoxanthine is also an indicator of hypoxia [35,36] and a marker of energy stress associated with intense physical training [31,32]. Furthermore, patients who received erythrocyte concentrates that are near the end of their shelf life have increased hypoxanthine concentrations of up to 100 µM [37].

### 3.2. Eicosanoids as Potential Quality Indicators in Serum

Our data indicate that several eicosanoids such as 8-(S)-HETE, 12-HETE, Tetranor-12-(S)-HETE, 15-(S)-HETE, 12-(S)-HEPE, and ±13-HODE increase in a time-dependent manner in human serum after a prolonged TTC. Eicosanoids are biologically active metabolites that are generated by enzymatic oxidation and rearrangement of free polyunsaturated fatty acids (PUFAs) such as arachidonic acid (AA), linolenic acid (LA), or eicosapentaenoic acid (EPA) through concerted action of cyclooxygenases (COX), lipoxygenase (LOX), and cytochrome P-450 (CYP), or via autoxidation [38,39]. These enzymes are strongly activated in platelets during aggregation and coagulation [40,41], induced by clot activators in serum samples immediately after blood collection. The previously observed high concentrations of eicosanoids in serum [42] compared to EDTA plasma results from the increased release and metabolism of PUFAs. For example, Dorow and colleagues found tremendously elevated levels of 11-, 12-, 15-HETE, 12-HHT, and TxB_2_ of up to 18,803% relative to EDTA plasma [43].

We show that some eicosanoids (8-(S)-HETE, 12-HETE, Tetranor-12-(S)-HETE, 15-(S)-HETE, 12-(S)-HEPE) are particularly suitable to identify samples with a TTC > 2 h with high sensitivity and specificity from both healthy individuals and patients with cardiovascular and rheumatic diseases (Table 1 and Table 2). However, the use of eicosanoids as QIs for delayed sample processing has clear limitations. For example, 12-HEPE and 12-HETE show significant changes (>2.77 × Coefficient of Variation) after two freeze-thaw cycles (FTC). Tetranor-12-HETE was strongly affected by 5 FTCs [43]. Additionally, the storage time at −80 °C can affect eicosanoid concentrations. Dorow and colleagues observed that eicosanoids such as 9-HODE or 13-HODE are unstable during storage for 6 months at −80 °C [43]. For this reason, the use individual eicosanoids as QI for retrospective analysis of already frozen serum samples should be discouraged. Instead, a combination of 8-(S)-HETE, 12-HETE, Tetranor-12-(S)-HETE, 15-(S)-HETE, 12-(S)-HEPE) should be used to estimate the TTC of these samples.

### 3.3. The Ratio of Ornithine to Arginine

In both serum and EDTA plasma, we observed a TTC-dependent increase in the ornithine/arginine ratio in samples kept at room temperature prior to centrifugation. This might be due to degradation of arginine to ornithine and urea by arginase, an enzyme highly expressed in erythrocytes [44]. Interestingly, the ability of the OA-ratio (Table 1 and Table 2) to discriminate TTC-dependent changes is not as specific and sensitive as predicted by Jain and colleagues [6]. This insufficient discriminatory power might be due to the age- and sex-associated differences between the *discovery* and the *validation samples*. While the original analysis was performed only with healthy male volunteers (*discovery sample*, n = 10), the validation has been done with samples of older female and male volunteers and patients. Saito and colleagues [45] have observed, that ornithine levels are associated with age and sex. In addition, patients with heart failure show increased arginine levels compared to healthy individuals [46].

Our study has several limitations. Dynamic changes in the composition of the serum and plasma metabolome are not exclusively altered by the pre-analytical phase, but are also influenced by several intrinsic and extrinsic factors, such as age and sex [45,47], the physiological rhythm [48], diet [49], exercise [50], fasting [51,52] or smoking status [53].

First, in this proof of concept study, the discovery samples were collected only from male participants with a limited age range compared to the validation cohorts, to focus entirely on the identification of potential quality indicators with as little influence as possible from strong inter-individual differences, for example, hormonal status. However, the successful validation of quality indicators in several heterogeneous validation cohorts points to the robustness of the identified QC indicators. Nevertheless, to avoid age-related bias, cohorts matched for age are recommended and should be used in future independent validation cohorts. Given that, for example, cardiovascular and rheumatic diseases are chronic age-related diseases that predominantly occur in the elderly, this might be challenging [54].

Second, for *validation samples 2–4*, 12-h fasting before blood collection could not be warranted, since samples were provided from pre-existing clinical biobank collections. Thus, several QC indicators from the group of amino acids, eicosanoids and nucleosides that were identified as quality indicators in the fasting *discovery sample*, may not have been successfully confirmed in the non-fasting *validation samples 2–4*.

Third, validation in this study was limited to two clinical entities, for example, patients with cardiovascular and rheumatic diseases. Thus, potential biomarker candidates, identified in this proof of concept study, which may serve as potential Qis, have to be prospectively validated in larger and independent sample cohorts, covering a wide range of clinical entities.

## 4. Materials and Methods

### 4.1. Study Design and Pre-Analytical Conditions

Forty-three self-reported healthy volunteers (*discovery sample* and *validation sample 1* and *2*), 60 cardiovascular patients and 79 rheumatic patients (*validation sample 3* and *4*) were included in this study (Appendix A). The demographic and clinical data are shown in Table 4.

Differing quantities (*validation sample 1*—106 mL, *validation sample 2*—33.1 mL) of blood were collected from each volunteer by peripheral venipuncture using a 21-gauge Safety-Multifly (Sarstedt AG & Co, KG, Nürmbrecht, Germany, www.sarstedt.com) blood collection system at room temperature. The volunteers were placed in an upright sitting position and the tourniquet released after blood flow started. We used 4.9 mL Serum-gel-S-Monovettes^®^ with a cloth activator (04.1935.001) and 2.7 mL and 4.9 mL K3EDTA-plasma-S-Monovettes^®^ (04.1917.001, 04.1931; Sarstedt AG & Co, KG, Nürmbrecht, Germany, www.sarstedt.com) for blood collection. EDTA plasma tubes were gently mixed by inverting the tubes five times immediately after blood collection. Serum tubes were kept for 30 min in an upright position to ensure a distinct separating layer after centrifugation (i.e., ‘sausage’ pattern). Blood collection tubes were centrifuged at 2500× *g* for 10 min at 20 °C (*discovery sample*, *validation sample 1, 2, 3* and *4*) and at 3309× *g* for 10 min at 20 °C (cardiovascular samples of *validation sample 3* and *4*). Blood collection tubes were handled at room temperature according to the conditions in Table 4. Collection time, pre- and post-centrifugation delay were monitored and recorded accordingly. All samples were divided into aliquots and were stored immediately after centrifugation at −80 °C in ultra-low freezers until shipping or further processing. Serum and EDTA plasma samples of all healthy volunteers (*discovery sample*, *validation sample 1* and *2*) were collected at the integrated biobank in Jena (IBBJ). Samples from *validation sample 3* and *4* were leftover materials from a clinical ward that were divided into aliquots after routine laboratory diagnostics from patients with rheumatic diseases. Samples kindly provided by the Interdisciplinary Bank of Biomaterials and Data in Würzburg (ibdw) were collected from patients suffering from cardiovascular diseases (*validation sample 3* and *4*). All samples from clinical wards were collected at random collection times.

### 4.2. Metabolite Profiling in Human Serum and EDTA Plasma

Serum and EDTA plasma samples with a pre-centrifugation delay of 0.5 h and 2 h were profiled by Metabolon Inc. (Durham, NC, USA). Sample handling, sample preparation, and metabolite profiling protocols were as described previously [55,56]. In brief, serum and EDTA plasma samples were prepared using the automated MicroLab STAR^®^ system (Hamilton Germany GmbH, Gräfelfing, Germany, www.hamiltoncompany.com), precipitated with methanol under vigorous shaking for 2 min (GenoGrinder2000, SPEX Sample Prep, Metuchen, NJ, USA, www.spexsampleprep.com) and then centrifuged. The supernatant was divided into five aliquots. These were subjected to four optimized methods for more hydrophilic, hydrophobic, polar, and nonpolar compounds and one aliquot was used as a backup. Chromatographic separation was achieved for all optimized methods using an ACQUITY (Waters Corporation, Milford, MA, USA) ultra-performance liquid chromatography (UPLC) system. The first method was performed using a Waters UPLC BEH C18 column (particle size 1.7 µm, column length 2.1 × 100 mm) and eluted by a gradient using a mixture of water and methanol, containing 0.05% perfluoropentanoic acid (PFPA) and 0.1% formic acid (FA). The second method utilized the same C18 column as the first but differed in the composition of elution solvents that were used for the gradient. Methanol, acetonitrile, and water, containing 0.05% PFPA and 0.1% FA were used at an overall higher organic content. The third method used a separate dedicated C18 column, and the elution was achieved by a gradient using methanol and water, containing 6.5 mM ammonium bicarbonate at pH 8. The fourth method was performed by using a HILIC column (Waters UPLC BEH Amide particle size 1.7 µm, column length 2.1 × 150 mm). A mixture of water and acetonitrile, containing 10 mM ammonium formate at pH 10.8, achieved the elution.

MS detection was performed using a Q-Exactive mass spectrometer (MS) equipped with a heated electrospray ionization source (Thermo Fisher Scientific, Waltham, MA, USA, www.thermofisher.com) operating in positive (first and second method) and negative (third and fourth method) ion mode. The Orbitrap mass analyzer operated at a mass resolution of 35.000 and alternated between MS and data-dependent MSn scans using dynamic exclusion. The scan range covered 70–1000 *m*/*z*.

Data extraction and peak identification were performed using Metabolon’s hardware and software [57,58]. Metabolites were identified using a reference library, which is based on authenticated standards that include retention time, accurate *m*/*z* ratios (±10 ppm) and associated MS/MS scores. These scores are based on the comparison between reference and experimental spectra of the ions of interest. This process is curated by visual inspection for quality control using software developed at Metabolon [55,56]. Peaks were quantified using the area-under-the-curve.

### 4.3. Quantitative Analysis of Nucleosides and Related Compounds

Sample analysis was carried out by MS-Omics (MSOmics, Vedbæk, Denmark, www.msomics.com) as follows. Proteins were precipitated using methanol, and the metabolites were extracted using a liquid–liquid extraction using chloroform and water. The aqueous phase was dried under nitrogen flow. Samples were reconstituted in eluent A (ultrapure water with 10 mM ammonium formate + 0.1% formic acid). The analysis was carried out using a UPLC system (UPLC ACQUITY, Waters) coupled with a time of flight mass spectrometer (Xevo G2 ToF, Waters). An electrospray ionization interface was used. The analysis was performed in negative and positive ionization modes. The UPLC was performed using a slightly modified version of the protocol described previously [59]. Data-processing used MZmine 2 [60] followed by curation using a custom made an in-house protocol. Identification of compounds was performed using both peak retention times (compared against authentic standards included in the analytical sequence) and accurate mass (with an acceptable deviation of 0.005 Da).

### 4.4. Quantitative Analysis of Eicosanoids

Serum and EDTA plasma samples were used to determine polyunsaturated fatty acids and eicosanoids. Sample handling, sample preparation, and metabolite measurements were as described previously [43,61]. In brief, 200 µL of serum or EDTA plasma were mixed with 450 µL precipitation solution containing the deuterium-labeled standard solution (c = 5 ng/mL in methanol-water 50:50 [*v*/*v*] and 50 ng/mL for AA-d8) under vigorous shaking for 2 min. Samples were centrifuged at 10,000× *g* for 5 min; the supernatant was transferred into a new vial and then stored at −80 °C until further analysis.

A Shimadzu UFLC LC-20A Prominence liquid chromatography system (Shimadzu Deutschland GmbH, Duisburg, Germany, www.shimadzu.de) was used. Online solid-phase extraction was achieved by a Strata-X column (particle size 25 µm; column length 20 × 2 mm i.d., Phenomenex, Aschaffenburg, Germany). Chromatographic separation of analytes was performed using a Kinetex C18 column (particle size 2.6 µm, column length 100 × 2.1 mm i.d., Phenomenex).

MS detection was performed using a 5500 QTrap mass spectrometer (AB Sciex LLC, Framingham, MA, USA) equipped with an electrospray ionization source in negative ionization mode. Scheduled multiple reaction monitoring experiments were used for the quantitative analysis of eicosanoids. In serum 14 eicosanoids and in EDTA plasma 16 eicosanoids could be analyzed.

### 4.5. Quantitative Analysis of Amino Acids

Calibration was performed using a physiological amino acid standard solved in 0.1 N HCl purchased from Laborservice Onken GmbH (product number: 5.403.151, Gründau, Germany). Amino acids had a concentration of 1 mmol/L, except for cystin (500 µmol/L), cystathionine (525 µmol/L), and urea (15 mmol/L). Additionally, alloisoleucin was solved in deionized water to a final concentration of 1 mmol/L. 100 µL of both stock solutions were combined, diluted using 300 µL of loading buffer and frozen at −80 °C until use. ClinCheck^®^ plasma control (level 1) for amino acids was used as internal quality control material (Recipe, Germany).

Next, 400 μL of sample material (serum and EDTA plasma) or 200 µL of controls were mixed with 100 μL/50 µL of a solution containing 10% sulfosalicylic acid to precipitate proteins. Samples were then put in the fridge at 2–8 °C for 0.5 h and then centrifuged at 16,060× *g* for 15 min. 100 μL of supernatant were diluted with 100 µL lithium loading buffer, at a pH 2.20 (product code: 80-2038-10) and vigorously mixed, and 30 µL were injected and measured.

The amino acids arginine and ornithine were measured using a BioChrom 30plus high-performance liquid chromatography cation-exchange system with ninhydrin detection (Biochrom Ltd., Cambridge, UK, www.biochrom.co.uk). For further details of the widely adopted method, see instructions for Biochrom 30+ Amino Acid Analyzers (2018) (Version 41 56 1783 IVD instruction for use English iss 16, http://www.biochrom.co.uk/user_downloads/?c=17). Amino acid concentrations were calculated offline using EZChrom Elite software (Version 3.3.1, Agilent, Santa Clara, CA, USA).

### 4.6. Boinformatics and Statistics for QI Discovery by Targeted Metabolite Profiling

Targeted metabolite profiling data were filtered, and missing values imputed using the MetImp 1.2 (https://metabolomics.cc.hawaii.edu/software/MetImp/, accessed on 30 June 2020) web tool. Filtering was applied by using the “modified 80% rule”, meaning that variables can be excluded from the data when the proportion of non-missing elements are accounted for less than 80% among each biological group. Missing values were imputed using the Gibbs sampler left-censored missing value imputation approach [62]. Profiling data were log10 transformed and compared using the paired *t*-test. To address multiple comparisons, we derived q-values [63] that extended the idea of the false discovery rates proposed by Benjamini and Hochberg [64].

Pathway enrichment analysis was performed using MetaboAnalyst 3.0 [65,66,67]. KEGG was selected as a reference pathway library. Altered metabolites in serum and EDTA plasma were uploaded to MetaboAnalyst 3.0 using their HMDB IDs and were analyzed using over-representation analysis (ORA) [68]. The selected ORA method was Fishers’ exact test. P-Values were adjusted for multiple testing using the Holm–Bonferroni method. We selected the node importance measure: out-degree centrality for topological analysis [69]. Additionally, we performed a chemical similarity enrichment analysis (ChemRICH –chemrich.fiehnlab.ucdavis.edu) and ontology mapping as described previously [17,70]. InChiKeys and SMILES codes were obtained from the PubChem identifier exchange service (https://pubchem.ncbi.nlm.nih.gov/idexchange/idexchange.cgi, accessed on 30 June 2020). Chemical enrichment statistics were calculated using the Kolmogorov–Smirnov test.

Data were also analyzed using Excel (Microsoft, Albuquerque, NM, USA) and OriginPro 2018 (Origin Lab Cooperation, Northampton, MA, USA).

### 4.7. Statistical Analyses in Validation Samples

In the case of multiple measurements for each healthy volunteer, we applied paired *t*-tests. Within one panel figure, all reported *p*-values were Bonferroni corrected for the number of comparisons. We generated receiver operating characteristic curves and areas under the ROC such that the curves contained only one measurement for each healthy volunteer—in the case of multiple measures, we randomly sampled from the measurements without replacement and generated the distribution via random draws. Data were analyzed in R version 3.6.2 [71] using the packages *tidyverse* [72], *caret* [73], *pROC* [74] and *ROCR* [75].

## 5. Conclusions

Controlling and assessing the quality and, thus, the fitness-for-purpose of samples is of significant importance for biobanks and research. However, frequently, biobanks are unable to completely control each pre-analytical step outside their direct area of responsibility and might receive samples with inadequately documented history. To overcome these shortcomings, reliable and versatile methods must be in place to ascertain the major pre-analytical factors affecting sample quality in order to achieve biobanking quality objectives. Hence, suitable methods and appropriate biomarkers, serving as QIs, must receive particular attention.

In this study, we have identified potential biomarker candidates, which may serve as QIs for pre-centrifugation delays. In particular, the HI-ratio, HG-ratio and the concentration of several eicosanoids are sensitive to a TTC longer than 1 or 2 h. For some of the selected QIs, we demonstrated high sensitivity and specificity for the discrimination of samples with different TTCs. While the QIs mentioned are promising, further independent prospective studies covering a wider spectrum of diseases are necessary to validate the proposed QIs and establish them for routine quality checks in biobanks.

## Figures and Tables

**Figure 1 metabolites-11-00638-f001:**
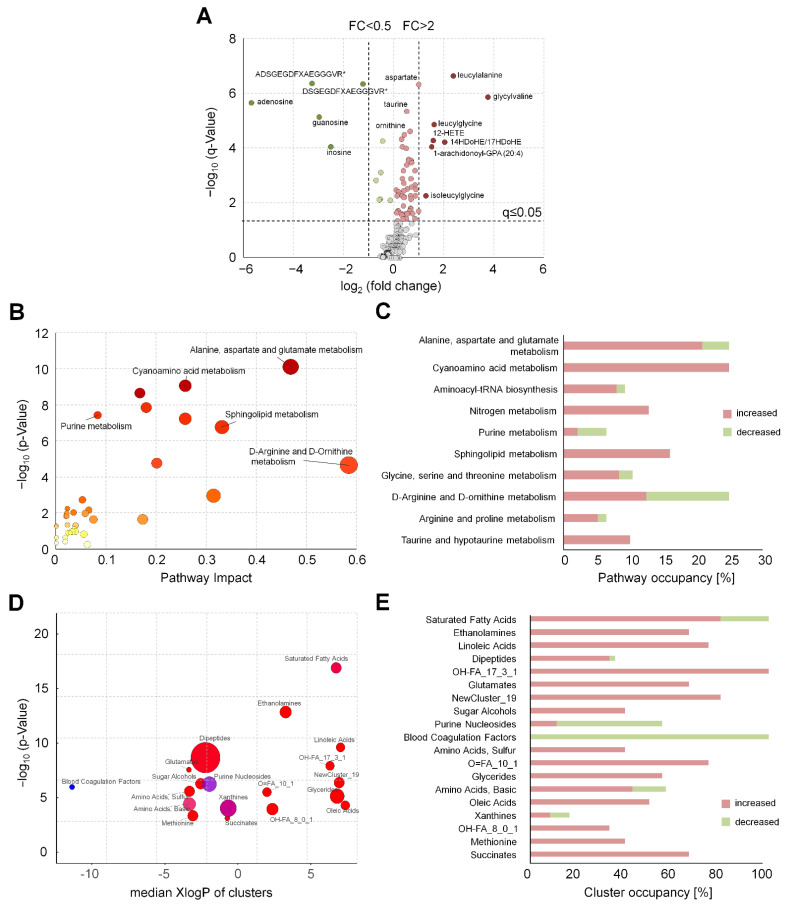
Chemical similarity and pathway enrichment analysis to identify novel QIs for the precentrifugation delay in human serum. (**A**) Volcano plot of serum metabolites visualizes the effects of prolonged blood incubation after 2 h before centrifugation. Red (increase) and green (decrease) circles indicate altered metabolite levels (*discovery sample*, n = 10). A paired *t*-test was performed, and we applied addressed multiple testing by deriving q-values to control the positive false discovery rate. (**B**) Represents the pathway enrichment analysis using the altered metabolites of (**A**). Node size represents the impact of the altered metabolites on the pathways and node color visualizes the q-value. (**D**) ChemRICH set enrichment statistics plot. Each node reflects an altered cluster of metabolites (*p*-value ≤ 0.05). Node sizes represent the total number of metabolites in each cluster set. Node color scale shows the proportion of increased (red) or decreased (blue) compounds in serum samples with a prolonged blood incubation of 2 h compared to the control at 0.5 h. Purple color nodes include both increased and decreased metabolites. (**C**) The pathway and (**E**) cluster occupancy visualize the ratio (in %) between increased and decreased metabolites within each altered pathway/cluster of (**A**).

**Figure 2 metabolites-11-00638-f002:**
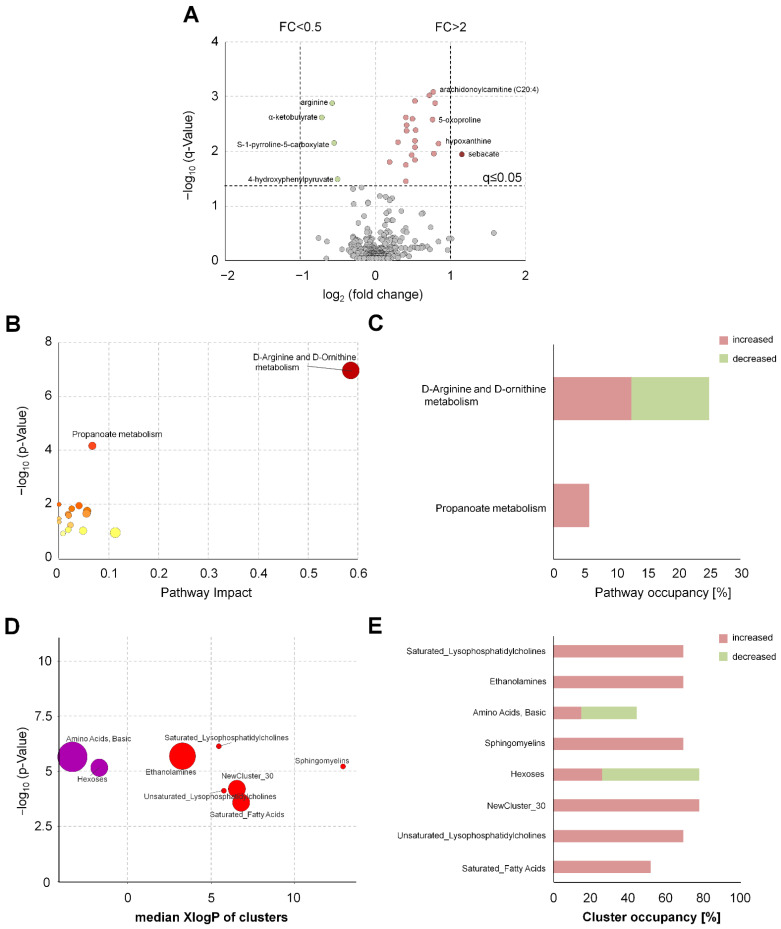
Chemical similarity and pathway enrichment analysis to identify novel QIs for the precentrifugation delay in human EDTA plasma. (**A**) Volcano plot of serum metabolites visualizes the effects of prolonged blood incubation after 2 h before centrifugation. Red (increase) and green (decrease) circles indicate altered metabolite levels (*discovery sample*, n = 10). A paired *t*-test was performed, and we applied addressed multiple testing by deriving q-values to control the positive false discovery rate. (**B**) Represents the pathway enrichment analysis using the altered metabolites of (**A**). Node size represents the impact of the altered metabolites on the pathways and node color visualizes the q-value. (**D**) ChemRICH set enrichment statistics plot. Each node reflects an altered cluster of metabolites (*p*-value ≤ 0.05). Node sizes represent the total number of metabolites in each cluster set. Node color scale shows the proportion of increased (red) or decreased (blue) compounds in EDTA plasma samples with a prolonged blood incubation of 2 h compared to the control at 0.5 h. Purple color nodes include both increased and decreased metabolites. (**C**) The pathway and (**E**) cluster occupancy visualize the ratio (in %) between increased and decreased metabolites within each altered pathway/cluster of (**A**).

**Figure 3 metabolites-11-00638-f003:**
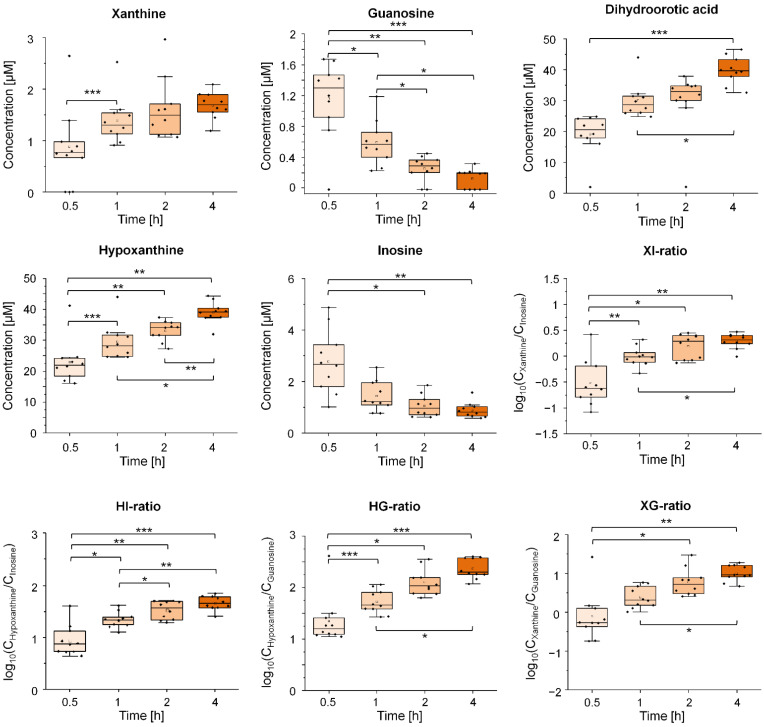
Impact of precentrifugation delay on selected purine and pyrimidine derivatives and their corresponding thresholds to verify selected quality indicators in human serum. Shown is the variation of the concentration of selected purine and pyrimidine derivatives in the serum of healthy volunteers after prolonged incubation for 0.5, 1, 2 and 4 h at room temperature (*validation sample 1*, n = 10). Additionally, diagnostic thresholds were calculated based on the identified pathway connections and the measured concentrations for the selected purine and pyrimidine derivatives. Asterisks indicate differences between the time points as indicated by the respective brackets (* *p* < 0.05, ** *p* < 0.01, *** *p* < 0.001). *p*-Values were calculated using linear mixed models or paired *t*-tests and were corrected for multiple testing using Bonferroni post-hoc test.

**Figure 4 metabolites-11-00638-f004:**
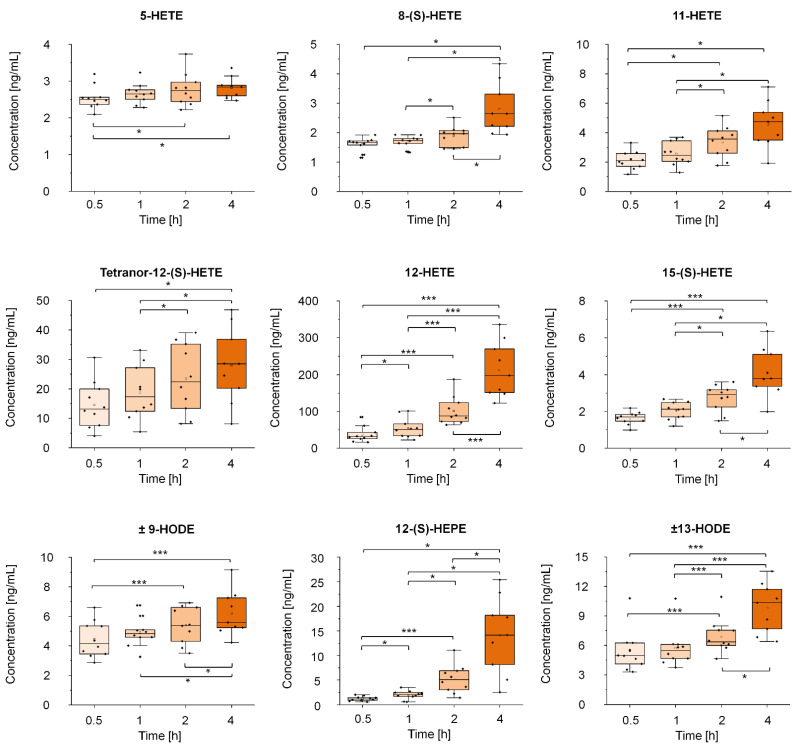
Influence of prolonged precentrifugation delay on selected eicosanoids in human serum. Boxplots visualize the variation of the concentration of selected eicosanoids in human serum of healthy volunteers after prolonged incubation for 0.5, 1, 2 and 4 h at room temperature (*validation sample 1*, n = 10) before centrifugation. Asterisks indicate differences between the time points as indicated by the respective brackets (* *p* < 0.05, *** *p* < 0.001). *p*-Values were calculated using linear mixed models or paired *t*-tests and were corrected for multiple testing using Bonferroni post-hoc test.

**Figure 5 metabolites-11-00638-f005:**
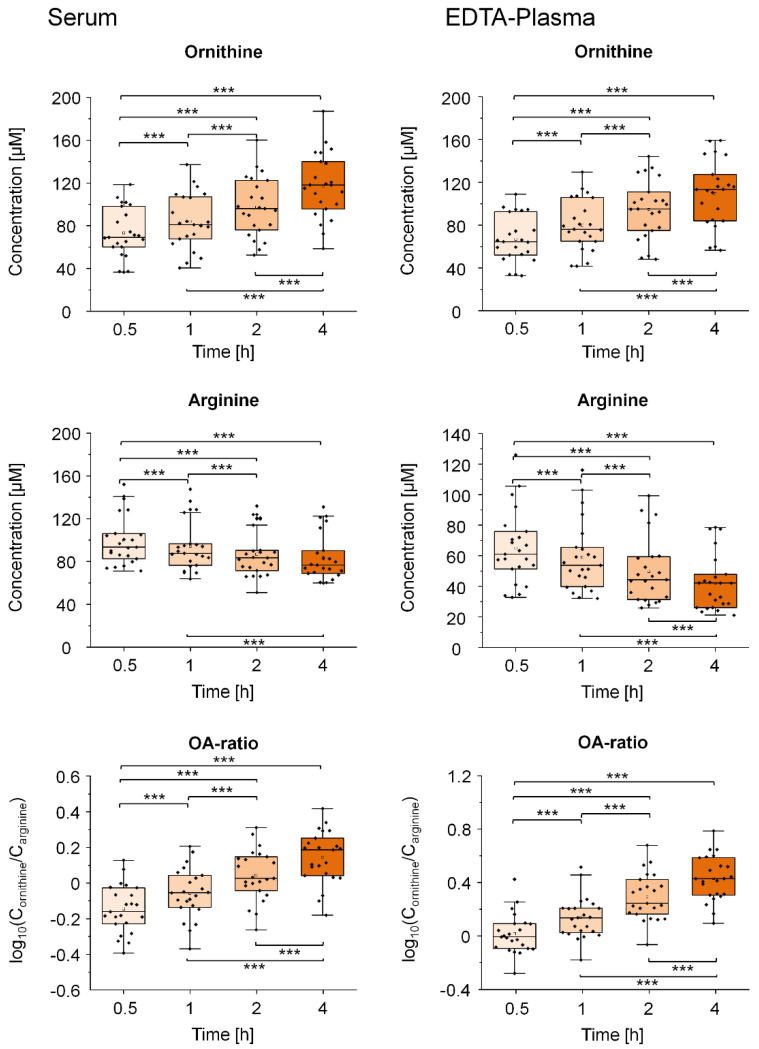
Impact of precentrifugation delay on arginine and ornithine in human serum and EDTA plasma. Boxplots display the variation of the concentration of arginine and ornithine in serum and EDTA plasma (*validation sample 2*, n = 23) of healthy volunteers after a prolonged incubation for 0, 0.5, 1, 2 and 4 h at room temperature before centrifugation. Additionally, diagnostic thresholds were calculated based on the identified pathway connections and the measured concentrations for both amino acids. Asterisks indicate differences between the time points as indicated by the respective brackets (*** *p* < 0.001). *p*-Values were calculated using linear mixed models or paired *t*-tests and corrected for multiple testing using Bonferroni post-hoc test.

**Table 1 metabolites-11-00638-t001:** Accuracy of all potential QIs for discrimination of TTC ≤ 1 h and 2 h in validation samples of healthy volunteers.

Potential QIs	Median Opt. Cutoff	Specificity n+/n > 1 h	Sensitivity 1-(n-/n) < 1 h	Median Opt. Cutoff	Specificity n+/n > 2 h	Sensitivity 1-(n-/n) < 2 h
Purines and Pyrimidines in serum of *validation sample 1*						
Guanosine ^#^	**0.50 µmol/L**	**100%**	**80%**	**0.28 µmol/L**	**100%**	**83%**
Hypoxanthine	32 µmol/L	75%	83%	**37 µmol/L**	**86%**	**80%**
Xanthine	1.5 µmol/L	75%	67%	1.6 µmol/L	80%	67%
Inosine ^#^	1.25 µmol/L	86%	75%	1.13 µmol/L	100%	67%
Dihydroorotic acid	32 µmol/L	83%	75%	38 µmol/L	94%	75%
XG-ratio	0.61	80%	80%	0.93	89%	75%
HI-ratio	**1.41**	**83%**	**86%**	1.57	78%	100%
HG-ratio	**2.06**	**100%**	**80%**	2.25	88%	75%
XI-ratio	0.14	78%	83%	0.24	73%	100%
Eicosanoids in serum of *validation sample 1*						
Tetranor-12(S)-HETE	-	-	-	0.45 ng/mL	88%	90%
12(S)-HEPE	-	-	-	**8.17 ng/mL**	**100%**	**100%**
(±)13-HODE	-	-	-	**6.83 ng/mL**	**83%**	**100%**
(±)9-HODE	-	-	-	5.31 ng/mL	62%	67%
15(S)-HETE	-	-	-	**3.36 ng/mL**	**100%**	**100%**
11-HETE	-	-	-	3.84 ng/mL	86%	75%
8(S)-HETE	-	-	-	**2.22 ng/mL**	**100%**	**100%**
12-HETE	-	-	-	**15.14 ng/mL**	**100%**	**100%**
5-HETE	-	-	-	2.63 ng/mL	60%	60%
12-oxo-ETE	-	-	-	**2.62 ng/mL**	**100%**	**100%**
Amino acids in serum of *validation sample 2*						
Arginine ^#^	86.15 µmol/L	64%	67%	83.4 µmol/L	70%	65%
Ornithine	91.24 µmol/L	67%	69%	106.31 µmol/L	74%	67%
OA-ratio	0.01	78%	77%	0.05	75%	75%
Purines and Pyrimidines in EDTA plasma of *validation sample 1*						
Hypoxanthine	12 µmol/L	50%	71%	11.5 µmol/L	80%	75%
Inosine ^#^	0.5 µmol/L	100%	33%	0.5 µmol/L	89%	0%
Dihydroorotic acid	12.2 µmol/L	50%	75%	12.65 µmol/L	67%	71%
HI-ratio	1.54	40%	75%	1.51	67%	80%
Eicosanoids in EDTA plasma of *validation sample 1*						
(±)13-HODE	-	-	-	5.45 ng/mL	78%	67%
(±)9-HODE	-	-	-	2.49 ng/mL	88%	67%
5-HETE	-	-	-	0.41 ng/mL	67%	50%
Amino acids in EDTA plasma of *validation sample 2*						
Arginine ^#^	51.24 µmol/L	73%	69%	46.62 µmol/L	75%	65%
Ornithine	87.94 µmol/L	71%	67%	110.38 µmol/L	88%	67%
OA-ratio	0.23	85%	73%	0.3	78%	83%

The table displays the calculated median optimal cutoffs for all identified potential QIs in *validation sample 1* and *2* in serum and EDTA plasma. ^#^ Cutoff values for arginine, inosine and guanosine (concentration decreases over time) are calculated for a TTC ≥ 1 or 2 h, respectively. QIs with the highest sensitivity and specificity are highlighted in **bold**.

**Table 2 metabolites-11-00638-t002:** Accuracy of all potential QIs for discrimination of TTC ≤ 1 h and 2 h in validation samples of patients with rheumatic and cardiovascular diseases.

Potential QIs	Median Opt. Cutoff	Specificity n+/n > 1 h	Sensitivity 1-(n-/n) < 1 h	Median Opt. Cutoff	Specificity n+/n > 2 h	Sensitivity 1-(n-/n) < 2 h
Purines and Pyrimidines in serum of *validation sample 3*						
Guanosine ^#^	0.50 µmol/L	43% (6/14)	96% (25/26)	0.28 µmol/L	65% (15/23)	71% (12/17)
Hypoxanthine	32 µmol/L	86% (12/14)	65% (17/26)	37 µmol/L	87% (20/23)	47% (8/17)
Xanthine	1.5 µmol/L	7% (1/14)	85% (22/26)	1.6 µmol/L	22% (5/23)	82% (15/26)
Inosine ^#^	1.25 µmol/L	57% (8/14)	88% (23/26)	1.13 µmol/L	43% (10/23)	88% (15/17)
Dihydroorotic acid	32 µmol/L	79% (11/14)	69% (18/26)	38 µmol/L	91% (21/23)	41% (7/17)
XG-ratio	0.61	36% (5/14)	96% (25/26)	0.93	70% (16/23)	71% (12/17)
HI-ratio	0.14	29% (4/14)	92% (24/26)	0.24	30% (7/23)	94% (16/17)
HG-ratio	**1.41**	**79% (11/14)**	**85% (22/26)**	1.57	70% (16/23)	82% (14/17)
XI-ratio	**2.06**	**100% (14/14)**	**73% (19/26)**	2.25	100% (23/23)	35% (6/17)
Eicosanoids in serum of *validation sample 3*						
Tetranor-12(S)-HETE	-	-	-	**0.45 ng/mL**	**92% (12/13)**	**100% (7/7)**
12(S)-HEPE	-	-	-	**8.17 ng/mL**	**100% (13/13)**	**86% (6/7)**
(±)13-HODE	-	-	-	6.83 ng/mL	62% (8/13)	71% (5/7)
(±)9-HODE	-	-	-	5.31 ng/mL	46% (6/13)	43% (3/7)
15(S)-HETE	-	-	-	3.36 ng/mL	85% (11/13)	71% (5/7)
11-HETE	-	-	-	3.84 ng/mL	77% (10/13)	14% (1/7)
8(S)-HETE	-	-	-	**2.22 ng/mL**	**100% (13/13)**	**86% (6/7)**
12-HETE	-	-	-	**15.14 ng/mL**	**92% (12/13)**	**86% (6/7)**
5-HETE	-	-	-	2.63 ng/mL	69% (9/13)	43% (3/7)
12-oxo-ETE	-	-	-	**2.62 ng/mL**	**100% (13/13)**	**86% (6/7)**
Amino acids in serum of *validation sample 4*						
Arginine ^#^	86.15 µmol/L	54% (14/26)	61% (27/44)	83.4 µmol/L	51% (24/47)	52% (12/23)
Ornithine	91.24 µmol/L	54% (10/26)	61% (27/44)	106.31 µmol/L	72% (34/47)	48% (11/23)
OA-ratio	0.01	46% (12/26)	73% (32/44)	0.05	51% (24/47)	70% (16/23)
Purines and Pyrimidines in EDTA plasma of *validation sample 3*						
Hypoxanthine	12 µmol/L	62% (8/13)	89% (24/27)	11.5 µmol/L	32% (9/28)	83% (10/12)
Inosine ^#^	0.5 µmol/L	23% (3/13)	89% (24/27)	0.5 µmol/L	11% (3/28)	75% (9/12)
Dihydroorotic acid	12.2 µmol/L	62% (8/13)	89% (24/27)	12.65 µmol/L	36% (10/28)	83% (10/12)
HI-ratio	**1.54**	**85% (11/13)**	**81% (22/27)**	1.51	43% (12/28)	75% (9/12)
Eicosanoids in EDTA plasma of *validation sample 3*						
(±)13-HODE	-	-	-	5.45 ng/mL	62% (8/13)	43% (4/7)
(±)9-HODE	-	-	-	2.49 ng/mL	38% (5/13)	29% (2/7)
5-HETE	-	-	-	0.41 ng/mL	54% (7/13)	43% (3/7)
Amino acids in EDTA plasma of *validation sample 4*						
Arginine ^#^	51.24 µmol/L	60% (3/5)	75% (33/44)	46.62 µmol/L	48% (11/23)	81% (21/26)
Ornithine	87.94 µmol/L	80% (4/5)	68% (30/44)	110.38 µmol/L	70% (16/23)	38% (10/26)
OA-ratio	0.23	60% (3/5)	84% (37/44)	0.3	43% (10/23)	69% (18/26)

The table displays the calculated median optimal cutoffs for all identified potential QIs in validation sample 1 and 2 in serum and EDTA plasma and applies them in validation sample 3 and 4. ^#^ Cutoff values for arginine, inosine and guanosine (concentration decreases over time) are calculated and applied for a TTC ≥ 1 or 2 h, respectively. n+, measurements with a positive test result; n-, measurements with a negative test result; n > 1 h, 2 h measurements with a TTC > 1 or 2 h; n < 1 h, 2 h, measurements with a TTC < 1 or 2 h; QIs with the highest sensitivity and specificity are highlighted in **bold**.

**Table 3 metabolites-11-00638-t003:** Proposed QIs to be considered for further investigation in serum and EDTA plasma.

Potential QIs in Serum	Median Opt. Cutoff	Specificity n+/n > 1 h	Sensitivity 1-(n-/n) < 1 h
HG-ratio	1.41	79% (11/14)	85% (22/26)
XI-ratio	2.06	100% (14/14)	73% (19/26)
		**Specificity** **n+/n > 2 h**	**Sensitivity** **1-(n-/n) < 2 h**
Tetranor-12(S)-HETE	0.45 ng/mL	92% (12/13)	100% (7/7)
12(S)-HEPE	8.17 ng/mL	100% (13/13)	86% (6/7)
8(S)-HETE	2.22 ng/mL	100% (13/13)	86% (6/7)
12-HETE	15.14 ng/mL	92% (12/13)	86% (6/7)
12-oxo-ETE	2.62 ng/mL	100% (13/13)	86% (6/7)
**Potential Qis in EDTA Plasma**	**Median Opt. Cutoff**	**Specificity** **n+/n > 1 h**	**Sensitivity** **1-(n-/n) < 1 h**
HI-ratio	1.54	85% (11/13)	81% (22/27)

The table displays the calculated median optimal cutoffs for all considered QIs in validation sample 1 and 2 in serum and EDTA plasma and applies them in validation sample 3 and 4. n+, measurements with a positive test result; n-, measurements with a negative test result; n > 1 h, 2 h measurements with a TTC > 1 or 2 h; n < 1 h, 2 h, measurements with a TTC < 1 or 2 h.

**Table 4 metabolites-11-00638-t004:** Demographic and clinical data of study cohorts.

**Characteristics**	**Data**	
*Discovery sample 1*		
No. of healthy volunteers	10	
Age (years) median, range	33 (28–38)	
Sex male/female	10/0	
Fasting before blood draw (12 h)	yes	
Non-smokers	yes	
Timepoints	TTC 0.5 and 2 h	
*Validation sample 1*		
No. of healthy volunteers	10	
Age (years) median, range	29 (21–54)	
Sex male/female	7/3	
Fasting before blood draw (12 h)	yes	
Non-smokers	yes	
Timepoints	TTC 0 (only EDTA plasma), 0.5, 1, 2 and 4 h	
*Validation sample 2*		
No. of healthy volunteers	23	
Age (years) median, range	43 (25–60)	
Sex male/female	6/17	
Fasting before blood draw (12 h)	no	
Non-smokers	yes	
Timepoints	TTC 0.5, 1, 2 and 4 h	
*Validation sample 3*		
No. of rheumatologic patients	20	
Age (years) median, range	70.5 (21–82)	
Sex male/female	14/6	
Disease	rheumatic	
Fasting before blood draw (12 h)	no	
*Validation sample 3*		
No. of cardilogic patients	20	
Age (years) median, range	64.5 (32–82)	
Sex male/female	10/10	
Disease	cardiovascular	
Fasting before blood draw (12 h)	no	
*Validation sample 4*		
No. of cardiologic patients	40	
Age (years) median, range	73 (48–89)	
Sex male/female	20/20	
Disease	cardiovascular	
Fasting before blood draw (12 h)	no	
*Validation sample 4*	Serum	EDTA plasma
No. of rheumatologic patients	30	49
Age (years) median, range	63 (32–88)	65 (20–90)
Sex male/female	14/16	23/26
Disease	rheumatic	rheumatic
Fasting before blood draw (12 h)	no	no

## Data Availability

The data presented in this study are available upon request from the corresponding authors. The data are not publicly available due to the privacy of the patients assisted in the research.

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
