# Peer review of "Metabolite Ratios as Quality Indicators for Pre-Analytical Variation in Serum and EDTA Plasma"

_metabolites, 2021, doi:10.3390/metabo11090638_

Round 1

Reviewer 1 Report

The manuiscript describes several new potential quality indicators (QI) reflecting delay in sample processing, time-to-centrifugation (TTC) in human serum and EDTA plasma  samples. The manuscript is generaly well written, but I have several concerns that should be addressed:

  1. There seem to be some differences between sample groups regarding sex of subjects, smoking status, fasting before taking the sample, time of collection. Those variations should be taken into account and commented in result interpretation.
  2. The validation samples 1 and 2, and 3 and 4 have different centrifugation speed. Possibly because they are from different labs/wards, but why wasn't that standardized before the study?
  3. Figures 1 and 2 B are not clear and nothing much is visible on them so they should be revorked or ommitted.
  4. Tables seem to be pretty large and cluttered, so perhaps it would be useful to have another small table with only significant result that would be usuful for further research (e.g. prpopsed qi indicators to be considered for firther studying with cutoffs, specificity, sensitivity..)

Author Response

Reviewer 1 

The manuscript describes several new potential quality indicators (QI) reflecting delay in sample processing, time-to-centrifugation (TTC) in human serum and EDTA plasma samples. The manuscript is generally well written, but I have several concerns that should be addressed: 

We thank the reviewer for the positive evaluation of our manuscript and for the helpful suggestions and comments.

1) There seem to be some differences between sample groups regarding sex of subjects, smoking status, fasting before taking the sample, time of collection. Those variations should be taken into account and commented in result interpretation.

We thank reviewer 1 for pointing this out. We have included a short paragraph that takes into account the limitations regarding the differences between sample groups in the discussion section.

2) The validation samples 1 and 2, and 3 and 4 have different centrifugation speed. Possibly because they are from different labs/wards, but why wasn't that standardized before the study?

First, of all we would like to thank the reviewer for addressing this point and apologize for any misunderstanding regarding the description of the centrifugation times. Only the cardiovascular samples from the biobank in Wuerzburg were centrifuged under different conditions. These samples are part of the validation samples 3 and 4. We have corrected the text accordingly.

The centrifugation speed was not standardized because these samples were selected for validation purposes from a preexisting collection from the biobank in Würzburg.

Additionally, we want to emphasize that both centrifugation speeds are commonly used in clinical laboratories and that all samples were aliquoted and frozen within a short time span (<30 min) to avoid TTF-dependent changes. Nevertheless, we cannot exclude the possibility that altered centrifugation conditions, might lead to variations in the subgroup of validation samples 3 and 4 by slightly reduced plasma platelet counts [1].

3) Figures 1 and 2 B are not clear and nothing much is visible on them so they should be revorked or ommitted.

Thank you for this important note regarding the visualization of the data in Figures 1B and 2B. We agree with reviewer 1 and we have omitted figure 1B and 2B.

4) Tables seem to be pretty large and cluttered, so perhaps it would be useful to have another small table with only significant result that would be usuful for further research (e.g. proposed qi indicators to be considered for firther studying with cutoffs, specificity, sensitivity..)

We thank reviewer 1 for this suggestion. We have included a short table 3 in the manuscript summarizing only the important results.

[1] Soderstrom, A.C.; Nybo, M.; Nielsen, C.; Vinholt, P.J. The effect of centrifugation speed and time on pre-analytical platelet activation. Clin Chem Lab Med 2016, 54, 1913-1920.

Reviewer 2 Report

This manuscript reports the preanalytical variations of selected purines and pyrimidines, eicosanoids and amino acids at different time-to-centrifugation (TTC) delays. The targets were first selected from a shotgun profiling of the serum metabolites. The breakdown pathways of the metabolites with the highest CV were selected for further study, e.g. xanthine/hypoxanthine, ornithine/arginine. Based on the concentration measured, the ROC curve was constructed to determine the optimal cutoff and performance statistics in discriminating TTC.

This project is well-designed and executed in a logically coherent manner. The data reported in the manuscript will serve as an important baseline for future method development. 

Author Response

We thank the reviewer for highlighting the quality of the present study.

Reviewer 3 Report

The submitted manuscript is a study of the effect of time-to-centrifugation (TTC) on the metabolome profile. Several metabolites were suggested as biomarkers of different TTCs. The study is well designed, and all conclusions are supported with the obtained results.

I have following comments and questions concerning the study:

1) page 2, line 49: As I could find, this is the first time where the explanation of TTF is met in the text, therefore the abbreviation should be placed here.

2) page 2, section 2.1: In some studies, Li heparin plasma is recommended for global metabolome profiling. Why was the УВЕФ plasma chosen for the study? Was any comparison performed in this study or described earlier in literature? There is no discussion on this question in the manuscript.

3) page 7, line 183: the bracket should be moved further in the text "...of (0, only..."

4) Fig. 4: I suggest using ng/mL as concentration units instead of pg/mL.

5) Fig. 5: I recommend to move apart the comparison brackets indicating the p-value <0.001 (***), as they look joined to each other.

6) page 11, table 1: I suggest using ng/mL instead of pg/mL.

7) page 14, line 329: There is an extra comma here ("by, Jane").

8) page 15, line 356: There is an extra comma here ("both, healthy").

9) page 15, Discussion as well as Conclusion: I think that some generalized conclusion is needed. It is not clear from the study, what are the recommendations on using biological samples, either fresh or taken from a biobank. What should an investigator decide - to use or not to use a sample after carrying out a metabolomics analysis of a sample and calculating the HI, OA, HG and other ratios?
If only 40-50 metabolites of 700+ ones were found to differ between the groups of 0.5 h TTC and 2h TTC, doesn't this mean that all other metabolites don't change their level and can be used for metabolomics screening?
Another important question is a comparison of the TTC influence on the metabolome profile with the time of storage. Are there any data where the stability of the same metabolites was studied for the samples stored at different temperatures and for different times?

10) page 15, table 3: In my opinion, the discovery sample 1 has several drawbacks, namely: only samples from male participants were used; the median age as well as the range of the participants age are quite different from those of other groups. I expect some rebuttal from the authors on this concern.

11) Also for table 3: How can authors comment that participants from the validation sets 2-4 were not fasting before blood sampling? A meal before blood sampling can influence the metabolomic profile.

12) page 21: Please provide DOI for each reference.

Author Response

Reviewer 3 

The submitted manuscript is a study of the effect of time-to-centrifugation (TTC) on the metabolome profile. Several metabolites were suggested as biomarkers of different TTCs. The study is well designed, and all conclusions are supported with the obtained results.

We would like to thank the reviewer for the positive evaluation of our study and the many helpful comments and very detailed suggestions for changes.

I have following comments and questions concerning the study:

1).page 2, line 49: As I could find, this is the first time where the explanation of TTF is met in the text, therefore the abbreviation should be placed here.

3).page 7, line 183: the bracket should be moved further in the text "...of (0, only..."

4).Fig. 4: I suggest using ng/mL as concentration units instead of pg/mL.

5).Fig. 5: I recommend to move apart the comparison brackets indicating the p-value <0.001 (***), as they look joined to each other.

6).page 11, table 1: I suggest using ng/mL instead of pg/mL.

7).page 14, line 329: There is an extra comma here ("by, Jane").

8).page 15, line 356: There is an extra comma here ("both, healthy").

We thank reviewer 3 for the careful revision of the manuscript. We changed all points above according to the suggestions.

2).page 2, section 2.1: In some studies, Li heparin plasma is recommended for global metabolome profiling. Why was the УВЕФ plasma chosen for the study? Was any comparison performed in this study or described earlier in literature? There is no discussion on this question in the manuscript.

We thank reviewer 3 for this important comment. EDTA plasma was chosen because, for example, about 40% of all liquid biospecimen in German biobanks are plasma, and a large percentage of these are EDTA plasma samples. Irrespective of the question of which anticoagulant is best suited for metabolome analyses, the initial aim of this study was to provide biobanks with an instrument that can be used to analyse the sample quality of EDTA plasma. Investigations for other plasma types and, in particular, comparisons between different plasma types are certainly also necessary, but this would go beyond the scope of this work. Furthermore, many studies prefer EDTA plasma [1-4] to perform metabolome analysis and some even suggest the use of EDTA plasma rather than li heparin [1,5]. We would like to ask for your understanding that this discussion should not be part of the manuscript from our point of view, as it would make the discussion too extensive and distract the reader from the main objectives of the study.

9).page 15, Discussion as well as Conclusion: I think that some generalized conclusion is needed. It is not clear from the study, what are the recommendations on using biological samples, either fresh or taken from a biobank. What should an investigator decide - to use or not to use a sample after carrying out a metabolomics analysis of a sample and calculating the HI, OA, HG and other ratios? If only 40-50 metabolites of 700+ ones were found to differ between the groups of 0.5 h TTC and 2h TTC, doesn't this mean that all other metabolites don't change their level and can be used for metabolomics screening?

We thank reviewer 3 for this comment. However, it is very difficult to give clear recommendations whether a sample should always be used fresh or whether metabolomics analyses can also be performed from frozen samples and which TTC should be followed in both cases. This depends primarily on the question of which metabolites are to be analyzed in a sample. Certainly, it is possible to measure the metabolites in EDTA plasma not affected by a prolonged TTC also in samples with a TTC of 2 hours. However, if unknown metabolites that are not among the 700+ tested are to be analyzed, a sample with a very short TTC should always be used. This is especially true for non-supervised screening approaches for metabolites. What an investigator should decide also depends heavily on the original research question. The quality indicators can be used to alert researchers and biobankers to inadequate sample quality and expected pre-analytical variability that can then be addressed. This is particularly important for existing sample collections with an inadequately documented history. Again, we would like to ask for your understanding that we feel this discussion would overload the manuscript.

Another important question is a comparison of the TTC influence on the metabolome profile with the time of storage. Are there any data where the stability of the same metabolites was studied for the samples stored at different temperatures and for different times?

Unfortunately, we are not aware of any studies (besides eicosanoids) on long-term storage at -80°C and the stability of nucleosides or the amino acids ornithine and arginine.

10) page 15, table 3: In my opinion, the discovery sample 1 has several drawbacks, namely: only samples from male participants were used; the median age as well as the range of the participants age are quite different from those of other groups. I expect some rebuttal from the authors on this concern.

We thank reviewer 1 for pointing this out. We agree that the discovery samples are limited in their variability. We selected this limited but well-defined cohort precisely because we initially wanted to focus entirely on identifying potential indicators of quality with as little influence as possible from strong interindividual differences, such as hormonal influences. Subsequently, we then included extra female participants as well as patients in the validation cohort to test the QC markers in a possibly real heterogeneous setting to validate their robustness. Nevertheless, we agree that to avoid age-related bias, cohorts matched for age are recommended and should be used in future independent validation cohorts. However, since cardiovascular and rheumatic diseases are chronic age-related diseases that predominantly occur in the elderly this might be challenging [6].

11) Also for table 3: How can authors comment that participants from the validation sets 2-4 were not fasting before blood sampling? A meal before blood sampling can influence the metabolomic profile.

We thank the reviewer for pointing our attention on this important matter. We added a short paragraph on these limitations in the discussion section.

12) page 21: Please provide DOI for each reference.

The references are according to metabolites instructions to authors’ format.

  1. Kamlage, B.; Maldonado, S.G.; Bethan, B.; Peter, E.; Schmitz, O.; Liebenberg, V.; Schatz, P. Quality markers addressing preanalytical variations of blood and plasma processing identified by broad and targeted metabolite profiling. Clinical Chemistry 2014, 60, 399-412.
  2. Malm, L.; Tybring, G.; Moritz, T.; Landin, B.; Galli, J. Metabolomic quality assessment of edta plasma and serum samples. Biopreserv Biobank 2016, 14, 416-423.
  3. Nishiumi, S.; Suzuki, M.; Kobayashi, T.; Yoshida, M. Differences in metabolite profiles caused by pre-analytical blood processing procedures. J Biosci Bioeng 2018, 125, 613-618.
  4. Yu, Z.H.; Kastenmuller, G.; He, Y.; Belcredi, P.; Moller, G.; Prehn, C.; Mendes, J.; Wahl, S.; Roemisch-Margl, W.; Ceglarek, U., et al. Differences between human plasma and serum metabolite profiles. Plos One 2011, 6.
  5. Gonzalez-Covarrubias, V.; Dane, A.; Hankemeier, T.; Vreeken, R.J. The influence of citrate, edta, and heparin anticoagulants to human plasma lc-ms lipidomic profiling. Metabolomics 2013, 9, 337-348.
  6. MacNee, W.; Rabinovich, R.A.; Choudhury, G. Ageing and the border between health and disease. Eur Respir J 2014, 44, 1332-1352.
